
# 1 Case Study: Risk Analysis by Overtopping During an Upstream
# 2 Landslide in Peñitas Dam, Mexico

Humberto J.F. Marengo[1], Alvaro A. Aldama[2]
[1]Engineering Institute, UNAM, Mexico.
[2] Independent Consultant, Cuernavaca, Mexico.
*Correspondence to*: Humberto Marengo (hmarengom@gmail.com)
**Abstract.** This research presents the procedure for risk assessment and reliability analysis to dam
overtopping (Peñitas) located downstream of a landslide dam. For the analysis are used six statistical
variables and their uncertainties, peak flood of the upstream dam, are evaluated with empirical formulas.
Highest water levels of the dam break event were computed using reservoir routing with an explicit
equation developed by authors. Afterward, overtopping risk analysis of Peñitas Dam was assessed for
different stages of excavation of the natural dam that were made for solve the problem. A sensitivity
analysis of duration of dam break is made, and also is calculated the possible upper elevation of Peñitas
dam, finding that is a recommended practice measurement in similar further cases. A methodology to do
an orderly and consistently analysis of risk is proposed to solve similar situations.
**1. Introduction.** Rain season on 2007 was very severe in the South-east part of Mexico and produced
during September and October higher flood until that date in Tabasco State. On 4 November 2007, took
place a landslide on Grijalva River, the second in the country with an extension of 80 Ha. Slide volume
was 55 million of cubic meters of rock and soil and made a natural dam upstream of Peñitas and
downstream of Malpaso dam´s. This landslide was 80m high, 800m length and 300m wide. Of total slide
volume, 15 million of cubic meters fell over the river and 40 million fell over the slopes of the river.
Hydroelectric Grijalva system (4 dams) with a total capacity of 37.50 hm$^3$ storage, was stopped in its
electrical production, in view that was not possible to take out the water go through the last dam, and the
reservoirs were completing full after the worst rainy season of history in that site. The landslide break
was very risky after a year with severe hydrological consequences in the Mexican southeast.




Potential failure of the natural dam meant a great risk over three million of people living downstream
Peñitas dam, and presented a high hazardous situation over Villahermosa, Cárdenas, Comalcalco and
Huimanguillo cities in Tabasco state.
Grijalva river in Mexico, has a mean annual runoff of 35,600 $hm^3$, and with Usumacinta River, total
runoff is 100 000 $hm^3$ (Rubio–Gutiérrez y Triana–Ramírez, 2006), the 77% of total Mexican runoff.
Grijalva river has an area of 60,256 $km^2$ (Dávila, 2011), begins in Guatemala, goes into Mexican territory
in Chiapas State, and flows toward Villahermosa, capital city of Tabasco State.
Over Grijalva river were build La Angostura (1975), Chicoasén (1980), Malpaso (1969) and Peñitas
(1987) dams.

**2. Landslide.** On November 4th 2007, at 20:32:00 local time, (02:32:00, GMT), happened the landslide
over the right margin of Grijalva river, 16 **km** upstream of Peñitas dam and 57 **km** downstream of
Malpaso dam , began with a detachment of a rock block of 1, 300 **m** length and 75 **m** thickness that fell
over de slope of the river hauling earth and rock that were constituted of limestone and sandstone rocks
that belong to the geological formations La Laja and Encanto of Oligocene–Miocene eras (Islas–Tenorio
et al., 2005). The sliding produced a natural dam over the river with approximate dimensions of 80m high,
800m length and 300m wide. Figure 1 (obtained from a Google map image in that moment, 2007).

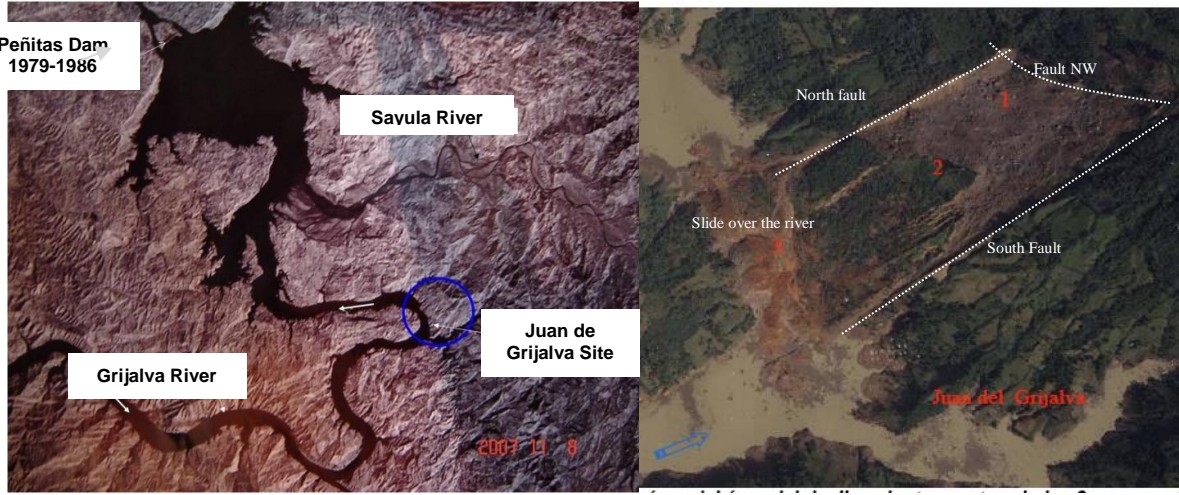


*Fig. 1. Landslide of Grijalva river* (obtained from © Google map image in that moment, 2007)*.*



**4. Geological framework.** Landslide was divided into three main blocks located by North and South faults, and in the upper part by a North-West fault. The characteristic of this formations is that they have limestone in the base, a stratigraphic formation that had a behavior like lubricant when received the intense precipitation (1450 mm) of the last days of October and November beginning.

Geological factors that produced the sliding were: (1) a tertiary sedimentary rock compound by limestone and sandstone rocks (2) stratigraphic units with dips between 8° y 10°, parallel to the slopes of the hillside, (3) high local relief, (4) inclined weak surfaces, (5) a high water table over the slide base, and (6), probably erosion of the hillside base occasioned by river erosion. This last factor produced a high increase of the pore pressure.

**4.1 Geological Model of the failure mechanism**

According with this, the landslide followed the next three steps:

a)      Before the separation of the natural slope, the rock mass had a safety factor of 1.5 according the scale of Carson y Kirkby (1972). However, after 5 days with intense precipitation, the rock mass with sandstones and limestones composition, began a slowly movement.

b)      The increase of pore pressure due to the saturation, occasioned a diminish of safety factor and the weight of the rock material, produced the sudden landslide due to the diminish of the shear resistance of the rock.

c)       The down portion of the rock, came into a heavy viscous mass, with debris and the rock mass movement was sudden producing a wave with 50m high that fell over a hamlet, dying 25 people, and obstructing totality the river.

The schematic situation presented during the first days of November, is shown in Fig. 2.




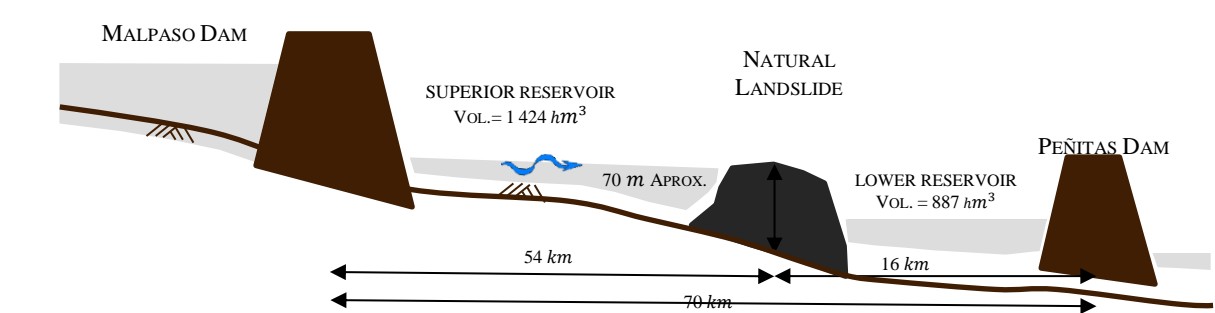

*Fig. 2. Landslide situation on Grijalva river.*

**5. Basis of the Study.** This research is made considering that engineers that front facing this kind of situations, need mathematical tools that let them to take decisions in order to measure possible situations of high risk, and they need to know alternatives for solve the problem.

Is usual in this kind of situations the tendency (like it was made in Peñitas), to excavate quickly trying to release storage water upstream and obviously diminish the risk as soon as possible. In the analysis herein presented, was studied the risk of failures for the highest level of storage water and during different stages of excavations made. Like process is made under pressure and must be promptly implemented, the authors decided use several empirical formulas for estimating the value of the peak flood. After this calculation then is showed risk analysis by overtopping using the advanced first-order second-moment (AFOSM) (Tang, 1984) method, that take into account several variables that intervene in the formulation.

This paper is organized as follows: first, a description of the landslide occurrence of November 2007 is made, is presented then the mathematical basis to get the behavior function, identifying explicit expression used for risk analysis model. This is followed by showing empirical formulas used to calculus of peak discharge in cases like this, and is presented the focus of the paper, which is risk analysis by overtopping of the practical actions taken for solving the situation during the emergency. The paper concludes by presenting the main lessons learned about risk analysis by overtopping due to the failure of the landslide and are showed practical applications that engineers can take in similar situations in the future.





**5. Background.** Many embankment structures, including dams built by humans, levees, dikes, barriers and natural dams formed by landslides, are located on rivers, lakes, and coastal shores around the world. Most of these structures play a very important role in flood defense, although many are also used for a water supply, power generation, transportation, and sediment retention. The limited safety levels of these structures are subject to natural deterioration. They may fail because of various trigger mechanisms (Costa, 1985; Foster, 2000; Allsop, 2007), including the high probability of failure under extreme conditions or when a natural event like a landslide occurs as is shown in this paper. These failures pose significant flood risks to people and property in the inundation area and cause a general disruption to society. A clear understanding of the predicting embankment failure processes is crucial for water infrastructure management.

Embankment breach formation by overtopping flood waters has been studied for many years, but recently it has been analyzed using complex two-dimensional depth-averaged flow models combined with soil erosion and slope failure algorithms by Froehlich (2004), Wang and Bowles (2006), and Faeh (2007). Models based on one-dimensional cross-section-averaged flow calculations combined with various sediment erosion and transport formulations have also been developed, including those by Ponce and Tsivolglou (1981), Nogeira (1984), Fread (1985), Al-Qaser (1991), Visser (1998), and Hanson et al (2005).

Froehlich et al (2008) states "*failure algorithms of a low levels of complexity are still needed when detailed simulations are not required or are not possible to work easily or conveniently. For these reasons, a simple empirical model that considers the formation of a breach in a presupposed way, usually growing into the shape of a trapezoid, is often applied in practice (United States Army Corps of Engineers, 1978)*".

Values of parameters used in such empirical breach formation models can be estimated using relations developed based on data collected from historic failures (United States Bureau of Reclamation, 1988; Froehlich, 1995; Mac Donald and Langridge-Monopolis, 1984; Wahl; 2004, and recently Machione I and II, 2008, and De Lorenzo, 2014). The uncertainties of parameters obtained in such a way can be large, as can their effects on planning actions are developed to minimize flood hazards. Such uncertainties may be quantified so that reasonable bounds on parameter values can be estimated and used to establish the





reliability of predicted dams outflow hydrographs at the dams, and the peak flow elevations and the flow rates at downstream locations of the system analyzed given by one-dimensional cross-section-average flow calculations.

**6. Approach to the Problem.** The analysis by overtopping is made under the following sequence: 1) is defined the flood routing over Peñitas dam with the development of an explicit expression developed by authors, that let estimating the level of water over the spillway, 2) are used for peak flood of landslide failure estimation several empirical methods, 3) is obtained the behavior function for the system that lets the assessment of the risk, 4) the methodology is applied to several excavation conditions that were defined in a practical form, but not were decided with a risk analysis tool, 5) the methodology is applied to practical cases like upper elevation of Peñitas dam.

**6.1 Flood Routing**

The reservoir routing follows continuity equation:

$$\frac{dS}{dt} = Q_l + Q_f + Q_s \tag{1}$$

where S is the storage in the reservoir of the Peñitas Dam, $Q_l$ is the flow generated by the landslide, $Q_f$, is the flow of tributaries rivers to the site of Peñitas, $Q_s$ is the flow extracted from the Peñitas spillway, and $t$ is the analysis time.

**6.2 Storage Capacity Curve**

Storage capacity elevation curve for the reservoir may be expressed as:

$$\frac{S-S_o}{S_F-S_o} = \left(\frac{Z-Z_o}{Z_F-Z_o}\right)^\alpha \tag{2}$$

where $Z$ is the elevation of the free water surface in the reservoir, $S_o$ is the storage corresponding to $Z_o$ elevation, which will be considered as a conservation level, $S_F$ is storage corresponding $Z_F$ elevation, which can be interpreted as the maximum level that can be reached when Eq. (1) is solved, $\alpha > 1$ is a regression constant.

From Eq. (2),

$$\frac{dS}{dt} = \alpha \frac{S_F-S_o}{Z_F-Z_o}\left(\frac{Z-Z_o}{Z_F-Z_o}\right)^{\alpha-1}\frac{dZ}{dt} = \alpha \frac{S_F-S_o}{Z_F-Z_o}\left(\frac{Z-Z_o}{Z_F-Z_o}\right)^{\alpha-1}\frac{dH}{dt} \tag{3}$$





where:

$$H = Z - Z_{cv} \tag{4}$$

is the spillway crest head and $Z_{cv}$ is crest elevation.

### 6.3 Hydrograph produced by the landslide

According to Fig. 3, the flow produced by the landslide can be written as

$$Q_l(t) = \begin{cases} 0, t \in (-\infty, 0) \\ Q_{pl}\left(1 - \dfrac{t}{t_{bl}}\right), t \in (0, t_{bl}) \\ 0, t \in (t_{bl}, \infty) \end{cases} \tag{5}$$

where $Q_{pl}$ is the peak flood and $t_{bl}$ is the base time of the hydrograph. It must be noted that the
triangular form of the hydrograph permits an increase in the volume if it is necessary.

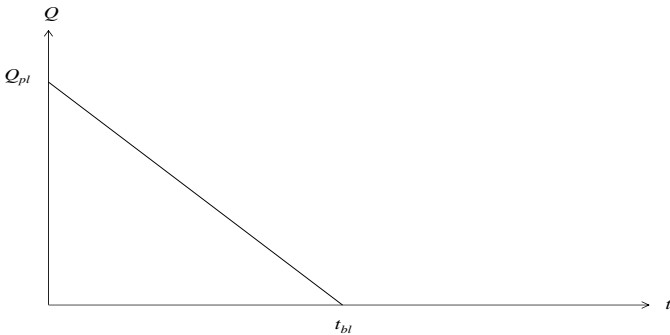

**Figure 3 Discharge law of the hydrograph**

### 6.4 Spillway discharge for the Peñitas Dam

The spillway discharge is shown in Fig. 4 and is given by

$$Q_s = \begin{cases} 0, \ H < H_o \\ CLH^{\frac{3}{2}}, \ H \geq H_o \end{cases} \tag{6}$$

where

$$H_o = Z_o - Z_{cv} \tag{7}$$

$C$ is the discharge coefficient, and L is the spillway length.




Note that if

$$Q_s < Q_l + Q_f, \quad t \in \left(0, t_{pf}\right) \tag{8}$$

then Eq. (6) may be written as

$$Q_s = \begin{cases} 0, & t < 0 \\ CLH^{\frac{3}{2}}, & t \geq 0 \end{cases} \tag{9}$$

In fact,

$$Q_{s,o} \equiv CLH_o^{3/2} \tag{10}$$

is the discharge in the spillway when $t=0$, as is shown in Fig.4.

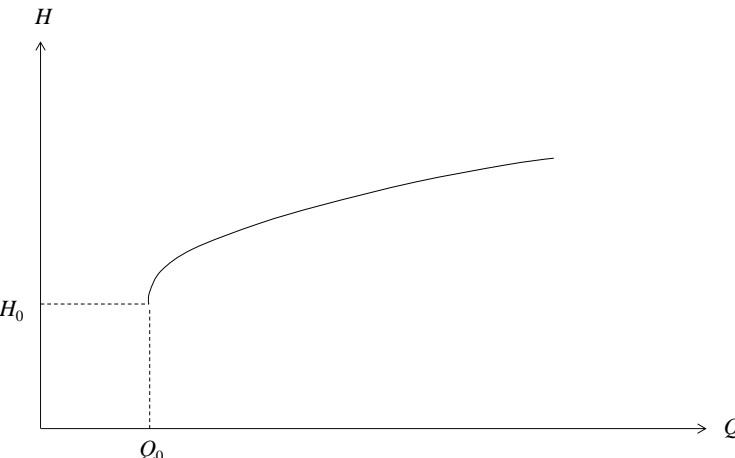

**Fig. 4. Discharge Law of the Spillway**

**6.5 Flood routing reviewed**
By substituting Eqs. (3) and (10) in Eq. (1),

$$F_c(H) = \alpha \frac{S_F - S_o}{Z_F - Z_o} \left(\frac{Z - Z_o}{Z_F - Z_o}\right)^{\alpha-1} \frac{dH}{dt} - \left[Q_l(t) + Q_f(t) - CLH^{\frac{3}{2}}\right] = 0, t > 0 \tag{11}$$

where $Q_l(t)$ y $Q_f(t)$ are given by Eqs. (5) and (6), and $F_c(\cdot)$ is a differential operator that acts over
the hydraulic head of the spillway, $H$.





## 6.6 Flood Routing Discretization

Eq. (13) has no analytical solution for an arbitrary value of α. Thus, a discretization solution based on the trapezoidal rule is done:

$$F_D = (H_j, H_{j+1}; \Delta t_{j+1/2}) \equiv \alpha \frac{S_F - S_o}{Z_F - Z_o} \left[ \frac{1}{2} \left( \frac{Z_c + H_j - Z_o}{Z_F - Z_o} \right)^{\alpha - 1} + \frac{1}{2} \left( \frac{Z_c + H_{j+1} - Z_o}{Z_F - Z_o} \right)^{\alpha - 1} \right] \frac{H_{j+1} - H_j}{\Delta t_{j+\frac{1}{2}}} -$$

$$\left[ \frac{Q_{l,j} + Q_{l,j+1}}{2} + \frac{Q_{f,j} + Q_{f,j+1}}{2} - \frac{CL}{2} \left( H_j^{3/2} + H_{j+1}^{3/2} \right) \right] = 0; \quad j = 0,1, \dots. \qquad (12)$$

where

$$H_j \approx H(t_j) \qquad\qquad (13)$$

$$H_{j+1} \approx H(t_{j+1}) \qquad\qquad (14)$$

Both are discrete approximations of the head values over the spillway crest in time $t_j$ and $t_{j+1}$. Thus,

$$Q_{l,j} = Q_l(t_j) \qquad\qquad (15)$$

$$Q_{l,j+1} = Q_l(t_{j+1}) \qquad\qquad (16)$$

$$Q_{f,j} = Q_f(t_j) \qquad\qquad (17)$$

$$Q_{f,j+1} = Q_f(t_{j+1}) \qquad\qquad (18)$$

In Eq. (14), we can use a time interval variable, defined as

$$\Delta t_{j+1/2} = t_{j+1} - t_j \qquad\qquad (19)$$

If $t_0=0$, Eq. (19) stay:

$$t_{j+1} = t_j + \Delta t_{j+\frac{1}{2}} = t_{j-1} + \Delta t_{j-\frac{1}{2}} + \Delta t_{j+\frac{1}{2}} = t_{j-2} + \Delta t_{j-\frac{3}{2}} + \Delta t_{j-\frac{1}{2}} + \Delta t_{j+\frac{1}{2}} = t_o + \sum_{k=0}^{j} \Delta t_{k+1/2} =$$

$$\sum_{k=0}^{j} \Delta t_{k+1/2} \, , j = 0,1, \dots. \qquad (20)$$

Finally, in Eq. (12), $F_D(\cdot;\cdot;\cdot)$ is a discrete operator that functionally depends on the heads $H_j$ and $H_{j+1}$ and from the parametric point of view, of the interval $\Delta t_{j+1/2}$.

It must also be observed that differences equation (12) is centered in $t_{j+1/2}=(t_j + t_{j+1})/2$, and it can be shown that building a continuum function twice differentiable around $H_j = H(t_j)$ that exactly satisfies Eq. (12), is possible to say:


$$F_D\left(H_j, H_{j+1}; \Delta t_{j+\frac{1}{2}}\right) = 0 \qquad (21)$$

Therefore, when differences equation (21) is solved, the differential modified equation
$F_C\left(H(t) + O\left(\Delta t^2_{j+\frac{1}{2}}\right)\right) = 0$ is being solved (Warming and Hyett. 1974). It must be noted that the
existence of $H(t)$ is guaranteed because the same can be built as a *cubic spline*.
Therefore, also is possible to show that Eq. (12) has a truncated error $T_{j+1/2} =$
$F_D[H(t_j), H(t_{j+1}); \Delta t_{j+1/2}] = O\left(\Delta t^2_{j+\frac{1}{2}}\right)$, (Smith, 1978)
Given that Eq. (12) defines an "ahead march" problem, this equation in finite differences is not lineal
in $H_{j+1}$ for known $H_j$, and then the analytical general solution for arbitrary values of α is not known.
With the objective of giving an analytical solution, a similar strategy to proposed by Beam and
Warming (1976) will be used that allows reaching an "implicit factorized scheme."
Remembering the Taylor theorem (Rosenlicht, 1968) for a function twice differentiable, $f = f(x)$ can
be written as
$$f(x + \Delta x) = f(x) + f'(x)\Delta x + \frac{1}{2}f''(\xi)\Delta x^2, \ x < \xi < x + \Delta x, \qquad (22)$$

where the residue has been written in a Lagrangian form.
By identifying $x$ with $H_j$ and $f(x)$ with $\left(\frac{Z_c + H_j - Z_o}{Z_F - Z_o}\right)^{\alpha-1}$, as well as $\Delta x$ with $H_{j+1} - H_j$, the Taylor
theorem (22) can be written as
$$\left(\frac{Z_c + H_{j+1} - Z_o}{Z_F - Z_o}\right)^{\alpha-1} = \left(\frac{Z_c + H_j - Z_o}{Z_F - Z_o}\right)^{\alpha-1} +$$

$$(\alpha - 1)\frac{(Z_c + H_j - Z_o)^{\alpha-2}}{(Z_F - Z_o)^{\alpha-1}}(H_{j+1} - H_j) + \frac{(\alpha-1)(\alpha-2)}{2}\frac{(Z_c + H_{j+\beta} - Z_o)^{\alpha-3}}{(Z_F - Z_o)^{\alpha-1}}(H_{j+1} - H_j)^2; \qquad (23)$$
$$0 < \beta < 1$$

Now identifying $x$ with $H_j$, $f(x)$ with $H_j^{3/2}$ and $\Delta x$ with $H_{j+1} - H_j$ for known $H_j$, it is possible again
to apply Taylor's theorem (22) as
$$H_{j+1}^{3/2} = H_j^{3/2} + \frac{3}{2}H_j^{1/2}(H_{j+1} - H_j) + \frac{3}{8}H_{j+1}^{-\frac{1}{2}}(H_{j+1} - H_j)^2; 0 < \gamma < 1 \qquad (24)$$



Obviously

$$H_{j+1} - H_j = O(\Delta t_{j+\frac{1}{2}}) \tag{25}$$

By substituting Eqs. (23) and (24) in Eq. (22) and considering the definition of differences $F_D$ given
in Eq. (12), then:

$$F_D = \left(H_j, H_{j+1}; \Delta t_{j+\frac{1}{2}}\right) \equiv \alpha \frac{S_F - S_o}{Z_F - Z_o} \left[\left(\frac{Z_c + H_j - Z_o}{Z_F - Z_o}\right)^{\alpha-1}\right] \frac{H_{j+1} - H_j}{\Delta t_{j+\frac{1}{2}}} - \left[\frac{Q_{l,j} + Q_{l,j+1}}{2} + \frac{Q_{f,j} + Q_{f,j+1}}{2} - \frac{CL}{2} H_j^{\frac{3}{2}} - \right.$$

$$\left. \frac{3}{4} CLH_j^{\frac{1}{2}} (H_{j+1} - H_j)\right] + O\left(\Delta t_{j+\frac{1}{2}}^2\right) = 0, \ j = 0,1,\dots.. \tag{26}$$

Thus, without altering the magnitude order of truncated error, i.e. of $O\left(\Delta t_{j+\frac{1}{2}}^2\right)$, from finite
differences of truncated given by Eq. (12), it is possible to build the next implicit scheme factorized
of second order for the approximate solution of differential equation of flood routing given by Eq.
(11), neglecting quadratic terms in $H_{j+1} - H_j$ and obviously in $\Delta t_{j+\frac{1}{2}}$ in Eq. (26):

$$F_D = \left(H_j, H_{j+1}; \Delta t_{j+\frac{1}{2}}\right) \equiv \alpha \frac{S_F - S_o}{Z_F - Z_o} \left[\left(\frac{Z_c + H_j - Z_o}{Z_F - Z_o}\right)^{\alpha-1}\right] \frac{H_{j+1} - H_j}{\Delta t_{j+\frac{1}{2}}} + \frac{3}{4} CLH_j^{\frac{1}{2}} H_{j+1} - \frac{1}{2} \left[Q_{l,j} + Q_{l,j+1} + Q_{f,j} + \right.$$

$$\left. Q_{f,j+1} - \frac{CL}{2} H_j^{\frac{3}{2}}\right)\right] = 0, j = 0,1\dots. \tag{27}$$

where

$$H_j \approx H(t_j) \tag{28}$$

$$H_{j+1} \approx H(t_{j+1}) \tag{29}$$

are discrete approximations of head values over the spillway crest that acquires in the times $t_j$ and
$t_{j+1}$. A truncated error can be shown that is given by Eq. (27):
$T_{j+1/2} = F_D\left(H(t_j), H(t_{j+1}); \Delta t_{j+\frac{1}{2}}\right) = O\left(\Delta t_{j+\frac{1}{2}}^2\right)$. The approximation order of Eq. (12) is not
affected; however, Eq. (26) can be written as




$\quad \alpha \frac{S_F-S_o}{Z_F-Z_o}\left[\left(\frac{Z_c+H_j-Z_o}{Z_F-Z_o}\right)^{\alpha-1}\right]H_{j+1} - \alpha \frac{S_F-S_o}{Z_F-Z_o}\left[\left(\frac{Z_c+H_j-Z_o}{Z_F-Z_o}\right)^{\alpha-1}\right]H_j + (\frac{3}{4}\Delta t_{j+\frac{1}{2}})CLH_j^{\frac{1}{2}}H_{j+1}-\frac{1}{2}\Delta t_{j+\frac{1}{2}}\left(Q_{l,j}+\right.$
$\qquad \left. Q_{l,j+1}+Q_{f,j}+Q_{f,j+1} - \frac{1}{2}CLH_j^{\frac{3}{2}}\right) = 0; j = 0,1,\dots\dots\dots$ (30)
and:
$\quad H_{j+1} = \dfrac{\alpha\frac{S_F-S_o}{Z_F-Z_o}\left[\left(\frac{Z_c+H_j-Z_o}{Z_F-Z_o}\right)^{\alpha-1}\right]H_j + \frac{1}{2}\Delta t_{j+\frac{1}{2}}\left(Q_{l,j}+Q_{l,j+1}+Q_{f,j}+Q_{f,j+1}-\frac{1}{2}CLH_j^{\frac{3}{2}}\right)}{\alpha\frac{S_F-S_o}{Z_F-Z_o}\left[\left(\frac{Z_c+H_j-Z_o}{Z_F-Z_o}\right)^{\alpha-1}\right]+(\frac{3}{4}\Delta t_{j+\frac{1}{2}})CLH_j^{\frac{1}{2}}}\quad j = 0,1,\dots$ (31)
Recursive Eq. (31) let the calculus of the flood routing over the Peñitas Reservoir and allows the
calculation of discharged flows by the spillway that correspond to each interval of time, given by
Eq. (31):
$\qquad\qquad\qquad Q_{s,j+1} \equiv CLH_{j+1}^{\frac{3}{2}}; j = 0,1,\dots$ (32)

It must be observed that with this analysis, associated to time design flood, must coincide with the
flood caused by the landslide, which is unlikely to happen. An analysis with different times in each
event is a motive for future research.

Maximum water elevation occurs once the landslide peak flow is reached and is given by equating
inflow and outflow discharges as is shown in Fig. 5, ($Q_1 \equiv Q_*$). In other words, the value $H_1 \equiv H_*$ is
given by Eq. (31), where the time is given by $t_1 \equiv t_*$, in Eq. (31):

$\qquad\qquad\qquad Q_* \equiv CLH_*^{\frac{3}{2}} = Q_{pf}\left(1 - \frac{t_*-t_{pf}}{t_{bf}-t_{pf}}\right)$ (33)

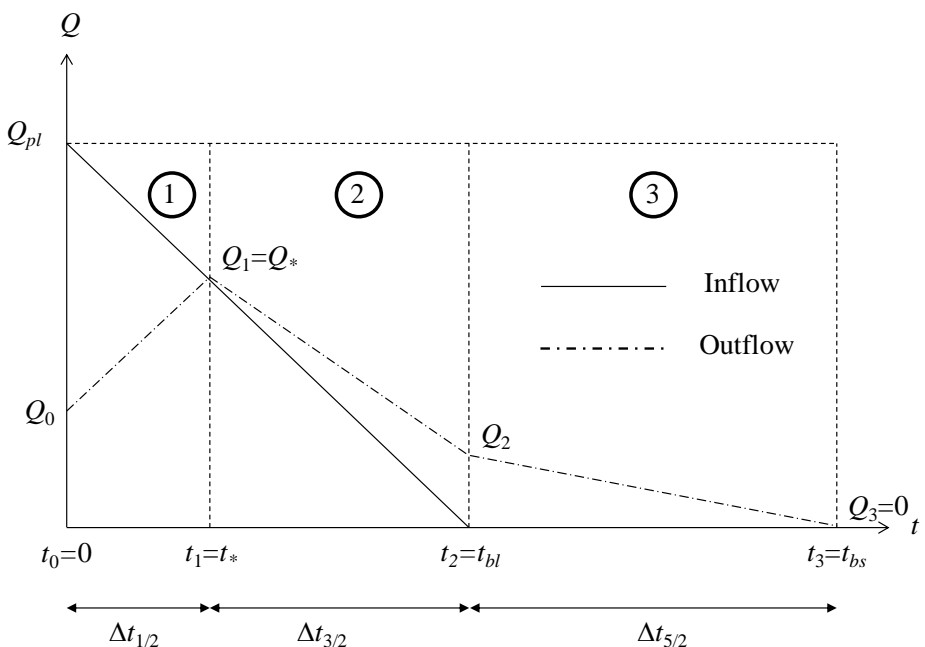

**Fig.5 Schematic representation of Inflow-Outflow to Peñitas River.**

## 6.7 Ordinary Risk Case

In the case that only the failure of the natural dam is present without floods from the tributaries, the analysis will be denominated "Ordinary Risk Case," then Eq. (31) continues being applicable with the consideration that $Q_{f,j}=Q_{f,j+1}\equiv 0$, $j=0,1,\dots$ . In this case, Fig. 5 shows that the maximum head belongs to $j=0$ and is given by:

$$H_{j+1} = \frac{\alpha\frac{S_F-S_o}{Z_F-Z_o}\left[\left(\frac{Z_C+H_0-Z_o}{Z_F-Z_o}\right)^{\alpha-1}\right]H_0 + \frac{1}{2}\Delta t_{\frac{1}{2}}\left(Q_{l,0}+Q_{l,1}-\frac{1}{2}CLH_0^{\frac{3}{2}}\right)}{\alpha\frac{S_F-S_o}{Z_F-Z_o}\left[\left(\frac{Z_C+H_0-Z_o}{Z_F-Z_o}\right)^{\alpha-1}\right]+(\frac{3}{4}\Delta t_{\frac{1}{2}})CLH_0^{\frac{1}{2}}} \quad j=0,1,\dots \qquad (34)$$

According with this Fig. 5,

$$Q_{l,0} = Q_{p,l} \qquad (35)$$

$$Q_{l,1} = \left(1 - \frac{t_*}{t_{bf}}\right)Q_{p,l} \qquad (36)$$

$$\Delta t_{1/2} = t_* \qquad (37)$$



By substituting Eqs. (35) through (37) in Eq. (34),

$$H_{j+1} = \frac{\alpha \frac{S_F - S_o}{Z_F - Z_o}\left[\left(\frac{Z_c + H_j - Z_o}{Z_F - Z_o}\right)^{\alpha-1}\right]H_0 + \frac{1}{2}t_*\left(\left(2 - \frac{t_*}{t_{bl}}\right)Q_{pl} - \frac{1}{2}CLH_0^{\frac{3}{2}}\right)}{\alpha \frac{S_F - S_o}{Z_F - Z_o}\left[\left(\frac{Z_c + H_0 - Z_o}{Z_F - Z_o}\right)^{\alpha-1}\right] + (\frac{3}{4}t_*)CLH_0^{\frac{1}{2}}} \quad j = 0,1,\dots \qquad (38)$$

Analogous to Eq. (32), equating inflow and outflow discharges, when $t=t_*$ (as in Fig. 4)

$$Q_* = CLH_*^{\frac{3}{2}} = \left(1 - \frac{t_*}{t_{bl}}\right)Q_{p,l} \qquad (39)$$

By substituting Eq. (38) in Eq. (39),

$$CL\left\{\frac{\frac{S_F - S_o}{Z_F - Z_o}\left[\left(\frac{Z_c + H_j - Z_o}{Z_F - Z_o}\right)^{\alpha-1}\right]H_0 + \frac{1}{2}t_*\left(2Q_{pl} - \frac{1}{2}CLH_0^{\frac{3}{2}}\right) - Q_{pl}\frac{t_*^2}{2t_{bl}}}{\frac{S_F - S_o}{Z_F - Z_o}\left[\left(\frac{Z_c + H_0 - Z_o}{Z_F - Z_o}\right)^{\alpha-1}\right] + (\frac{3}{4}t_*)CLH_0^{\frac{1}{2}}}\right\}^{3/2} = \left(1 - \frac{t_*}{t_{bl}}\right)Q_{p,l} \qquad (40)$$

Equation (40) is not linear in $t_*$ and can be expressed as a polynomial equation of sixth degree. By
the Abel impossibility theorem, it is not possible obtain an explicit solution; therefore, an
alternative method is proposed as the one used before for determining $t_*$. Let now

$$A = \alpha \frac{S_F - S_o}{Z_F - Z_o}\left[\left(\frac{Z_c + H_j - Z_o}{Z_F - Z_o}\right)^{\alpha-1}\right]H_0 \qquad (41)$$


$$B = \frac{1}{2}\left(2Q_{pl} - \frac{1}{2}CLH_0^{\frac{3}{2}}\right) \qquad (42)$$


$$D = \alpha \frac{S_F - S_o}{Z_F - Z_o}\left[\left(\frac{Z_c + H_j - Z_o}{Z_F - Z_o}\right)^{\alpha-1}\right] \qquad (43)$$


$$E = \frac{3}{4}CLH_0^{\frac{1}{2}} \qquad (44)$$


By expanding the left member of Eq. (40) in Taylor series, we have (as in Eqs. (38) and (39) through

302    (44)):


$$\left\{\frac{\frac{S_F - S_o}{Z_F - Z_o}\left[\left(\frac{Z_c + H_j - Z_o}{Z_F - Z_o}\right)^{\alpha-1}\right]H_0 + \frac{1}{2}t_*\left(2Q_{pl} - \frac{1}{2}CLH_0^{\frac{3}{2}}\right) - Q_{pl}\frac{t_*^2}{2t_{bl}}}{\frac{S_F - S_o}{Z_F - Z_o}\left[\left(\frac{Z_c + H_0 - Z_o}{Z_F - Z_o}\right)^{\alpha-1}\right] + (\frac{3}{4}t_*)CLH_0^{\frac{1}{2}}}\right\}^{\frac{3}{2}} = \left(\frac{A + Bt_* + B't_*^2}{D + Et_*}\right)^{3/2} = \left(\frac{A}{D}\right)^{3/2} + \frac{3}{2}\left(\frac{A}{D}\right)^{1/2}\frac{BD - AE}{D^2}t_* +$$


$$O(\Delta t_{\frac{1}{2}}^2) \qquad (45)$$





By neglecting the terms of $O(\Delta t_{\frac{1}{2}}^2)$ in this equation, by substituting the result in Eq. (39) and by
solving for $t_*$, we have
$$t_* = \frac{Q_{pl} - CL\left(\frac{A}{D}\right)^{3/2}}{\frac{3}{2}CL\left(\frac{A}{D}\right)^{1/2}\left(\frac{B}{D} - \frac{AE}{D^2}\right) + \frac{Q_{pl}}{t_{bl}}} \tag{46}$$

From Eqs. (41) through (44), we have
$$\frac{A}{D} = H_0 \tag{47}$$

$$\frac{B}{D} = \frac{Q_{pl} - \frac{1}{4}CL\left(\frac{A}{D}\right)^{3/2}}{\frac{S_F - S_o}{Z_F - Z_o}\left[\left(\frac{Z_c + H_0 - Z_o}{Z_F - Z_o}\right)^{\alpha - 1}\right]} \tag{48}$$

$$\frac{E}{D} = \frac{3}{4}\frac{CL(H_0)^{1/2}}{\alpha\frac{S_F - S_o}{Z_F - Z_o}\left[\left(\frac{Z_c + H_0 - Z_o}{Z_F - Z_o}\right)^{\alpha - 1}\right]} \tag{49}$$

Hence,
$$\frac{B}{D} - \frac{AE}{D^2} = \frac{Q_{pl} - CLH_0^{3/2}}{\alpha\frac{S_F - S_o}{Z_F - Z_o}\left[\left(\frac{Z_c + H_0 - Z_o}{Z_F - Z_o}\right)^{\alpha - 1}\right]} \tag{50}$$

By substituting Eqs. (47) through (50) in Eq. (45),

$$t_* = \frac{Q_{pl} - CLH_0^{3/2}}{\frac{3}{2}CLH_0^{1/2}\frac{Q_{pl} - CLH_0^{3/2}}{\alpha\frac{S_F - S_o}{Z_F - Z_o}\left[\left(\frac{Z_c + H_0 - Z_o}{Z_F - Z_o}\right)^{\alpha - 1}\right]} + \frac{Q_{pl}}{t_{bl}}} \tag{51}$$

By finally substituting Eq. (51) in Eq. (38), the explicit expression for the maximum head is
obtained:



$$H_* =$$

$$\frac{\left\{\alpha\frac{S_F-S_o}{Z_F-Z_O}\left[\left(\frac{Z_c+H_j-Z_O}{Z_F-Z_O}\right)^{\alpha-1}\right]H_0+\frac{1}{2}\left[\cfrac{Q_{pl}-CLH_0^{\frac{3}{2}}}{\frac{3}{2}CLH_0^{1/2}\left[\cfrac{Q_{pl}-CLH_0^{3/2}}{\alpha\frac{S_F-S_o}{Z_F-Z_O}\left[\left(\frac{Z_c+H_0-Z_O}{Z_F-Z_O}\right)^{\alpha-1}\right]}+\frac{Q_{pl}}{t_{bl}}\right]}\left\|2Q_{pl}-\cfrac{Q_{pl}}{t_{bl}}\left[\cfrac{\left[\left(Q_{pl}-CLH_0^{\frac{3}{2}}\right)-\frac{1}{2}CLH_0^{\frac{3}{2}}\right]}{\frac{Q_{pl}-CLH_0^{3/2}}{\alpha\frac{S_F-S_o}{Z_F-Z_O}\left[\left(\frac{Z_c+H_0-Z_O}{Z_F-Z_O}\right)^{\alpha-1}\right]}+\frac{Q_{pl}}{t_{bl}}}\right]\right\|\right]\right\}}{\frac{S_F-S_o}{Z_F-Z_O}\left[\left(\frac{Z_c+H_0-Z_O}{Z_F-Z_O}\right)^{\alpha-1}\right]+\left(\frac{3}{4}\right)CLH_0^{\frac{1}{2}}\cfrac{Q_{pl}-CLH_0^{\frac{3}{2}}}{\frac{3}{2}CLH_0^{\frac{1}{2}}\cfrac{Q_{pl}-CLH_0^{\frac{3}{2}}}{\alpha\frac{S_F-S_o}{Z_F-Z_O}\left[\left(\frac{Z_c+H_0-Z_O}{Z_F-Z_O}\right)^{\alpha-1}\right]}+\frac{Q_{pl}}{t_{bl}}}} \quad \ldots(52)$$


## 7. Case Study

### 7.1 Water Level Upstream elevations of Landslide

Risk analysis was made considering that volumes and heights of water stored upstream are like shown in Fig.6. The minimum operation level in the Peñitas Dam is 85.00 masl (meters above sea level), that corresponds to the minimum operation level of the dam.

328
329

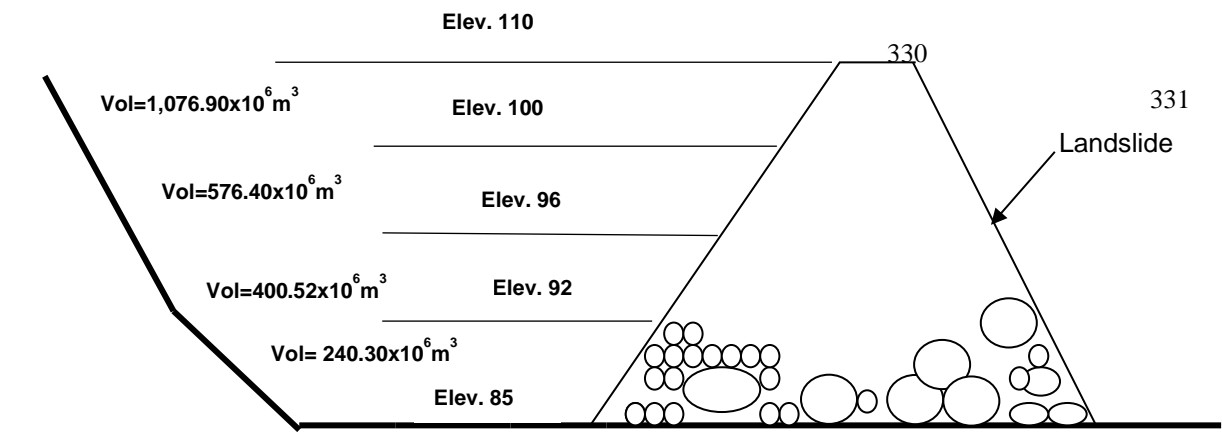

**Fig 6. Water Level Conditions in Landslide**

Under this situation, the highest risk correspond obviously to the maximum elevation (Elev. 110), and so successively, depending of excavation stage in progress, risk conditions diminished, but they were present in the site until elevation 92.00 masl was reached, because the water storage upstream under this elevation




could be stored in the reservoir between the landslide and Peñitas and in case of failures, the spillway has
the capacity of discharge it with none risk.
**7.2 Empirical Peak Flow Estimations of Dam Failure**
Many empirical formulations have been developed for predicting dam breach characteristics and peak
outflows, in general, empirical equation takes the form Eq. (1), and allow estimating the maximum peak
flow of the break dam considering the volume upstream $V_W$ in the moment of the failure ($m^3$), the height
of the dam $H_W$(m) and three correlation parameters a, b and c:
$$Q_{CA} = a\, V_W^b\ H_W^c \tag{53}$$

Drexel University (2006) did a compilation of several equations that are shown in Table 1.
**Table 1. Empirical Equations for estimating Peak Flow, (Drexel University, 2006).**

| Name | Equation | Coments |
|---|---|---|
| Hagen (1982) | $Q_p = 1.205(V_w H_w)^{0.48}$ (54) | Analized 18 failure dams by overtopping. |
| Costa (a) | $Q_p = 2.63(V_w H_w)^{0.44}$ (55) | Analized 31 failure dams with Hw in a range of 1.8m to 83.8m and Vw from 0.038 to 7 millons of $m^3$. |
| Mc Donald & Langridge (1984) (a) | $Q_p = 3.85(V_w H_w)^{0.411}$ (56) | Analized 42 failure dams with Hw in a range of 6m to 93m and Vw form 0.1 to 310 millons of $m^3$. |
| Costa (b) | $Q_p = 0.981(V_w H_w)^{0.42}$ (57) | |
| Mc Donald & Langridge (1984) (b) | $Q_p = 1.154(V_w H_w)^{0.411}$ (58) | |
| Froehlich (1995) | $Q_p = 0.607(V_w^{0.295} H_w^{1.24})$ (59) | Analized 22 failure dams with Hw in a range of 3.4m to 77.4m and Vw form 0.1 to 310 millons of $m^3$. |
| De Lorenzo (2014) | $Q_p = Q^* 0.228\alpha_0^{0.41} G^{0.531}$ <br> $Q_p = Q^* 0.228\alpha_0^{0.41}\left[\nu_e \dfrac{W_M}{\sqrt{g}Z_M^{7/2}}\right]^{0.531}$ <br> $Q_p = 0.1548\,(V_w^{0.531} H_w^{0.6415})$ (60) | $G = \dfrac{\nu_e}{V_w}$; $\nu_e = 0.07\dfrac{m}{s}$; $V_w = \dfrac{g^{1/2}H_w^{7/2}}{V_w}$ <br> $V_w$ is the volume in the reservoir and $H_w$ the height of Fig. 5. |





| | | | | | Eq. (60) was obtained substituting values by De Lorenzo for this analysis. (De Lorenzo, Macchione. 2014). |
|---|---|---|---|---|---|


Table 2 shows the computed discharges $Q_p$ obtained using these equations where the values of volume
and height of Fig. 6 are considered.

**Table 2. Peak Flow in Peñitas dam.**

| Upstream Elevation | $V_w$ ($10^6$ m$^3$) | $H_w$ (m) | Hagen (Eq. 54) | Costa (a) (Eq. 55) | Mc Donald (a) (Eq.56) | Costa (b) (Eq. 57) | Mc Donald (b) (Eq. 58) | Frohelich (Eq. 59) | De Lorenzo (Eq. 60) |
|---|---|---|---|---|---|---|---|---|---|
| 110.00 | 1076.9 | 25 | 121,322 | 101,746 | 74,085 | 23,568 | 22,878 | 15,174 | 75,819 |
| 100.00 | 576.4 | 15 | 70,369 | 61,898 | 46,680 | 14,626 | 14,004 | 6,698 | 39,460 |
| 96.00 | 450.52 | 11 | 51,164 | 46,195 | 35,409 | 11,019 | 10,614 | 4,095 | 26,657 |
| 92.00 | 240.30 | 7 | 32,314 | 30,153 | 23,839 | 7,354 | 7,145 | 2,011 | 15,208 |


By analyzing each equation, can be seen that Hagen and Costa (a) have the highest values with 121,322
m$^3$/s and 101,746 m$^3$/s, respectively; furthermore, Mc Donald (a) with 74,085 m$^3$/s and De Lorenzo with
75,819 m$^3$/s reach similar values. Costa (b) is 23,568 m$^3$/s, Mc Donald (b) is 22,878 m$^3$/s, and finally
Froehlich is 15,174 m$^3$/s.
For the risk analysis process, the equations of Hagen, Costa (a), Mc Donald (a) and De Lorenzo were
chosen.
**7.3 Landslide Duration**
From Froehlich (2008), the analysis of 74 cases of failures of earth dams were reported with failure modes
of (O): Overtopping, (P): Piping and (S): Sliding. Durations observed for the peak flow were very reduced
in the case of overtopping because dams with very reduced volume upstream were studied; in some cases,
the height of water was as the studied herein. The only cases with large volumes and heights were the
failure of the Oros Dam in Brazil ($V_w$=660 x$10^6$ m$^3$, and $H_w$=35.5m), in which the formation breach time



was 8.5 hours, and the Teton Dam that failed by piping ($V_w$=310 x$10^6$ m$^3$, and $H_w$=86.9m) with a breach

formation time of 1.25 hours.

This also was studied by Weiming (2011) *et al.* "*Based on experience, the embankment formation from its initiation to the final breach geometry can vary from a dozen minutes to a few hours. Engineers need a quick prediction of the breach flood to make timely warnings and decisions on evacuation and mitigation*."

**7.4 Analysis of Statistical Variables**

In this study, the uncertainty factors considered when analyzing the failure of the landslide are as follows:

1) Water volume in the upstream reservoir $V_w$; the uncertainty of volume is a reduction of the known capacity of upstream reservoir because with time there has been sedimentation; the standard deviation adopted is $\sigma_{vw}$=269.22x$10^6$ m$^3$ that corresponds to a coefficient of variation C.O.V.= 0.25.

2) Height of the upstream reservoir $H_w$, the situation is similar, and a great variation is expected; the standard deviation adopted was $\sigma_{Hw}$ =7.50 m with a variation coefficient C.O.V.= 0.30.

3) The discharge coefficient C of the spillway, in metric units, usually is 2 and is considered a standard deviation of $\sigma_c$ =0.14 that corresponds to a C.O.V.=0.070.

4) The length of the spillway L is a fixed geometric value, and the standard deviation taken is small $\sigma_L$ =1.40 m with a C.O.V.=0.0121.

5) The water variation of the Peñitas Reservoir $H_{i+1}$ is calculated with the variation of levels that exist between the minimum operation level (85.00 masl) and the crown elevation of the dam (98.00 masl). This new "shorted reservoir" also has received sedimentation over time; the standard deviation adopted was $\sigma_{Hi+1}$ =3.893m, with a C.O.V.=0.5561.

6) The time duration of the flood landslide adopted has a standard deviation $\sigma_{tbl}$=0.10h and a C.O.V.=0.05.

Statistical properties of the six uncertainty factors are summarized in Table 3.

**7.5 Analysis Considerations**

The failure mechanism analyzed is the ordinary risk case that is a sudden failure that occurs by piping and regressive erosion as established by De Lorenzo (ASCE, 2014) *et al.*: "*the relatively small size of the*





*initial hole usually allows discharges that are small in comparison with the expected peak discharge.*
*Moreover, during this stage the volume is usually negligible. The outflow grows considerably during the*
*stage in which the top of the pipe collapses into the breach. From this moment onwards, the failure can*
*be treated, as in the case of overtopping."*

**Table 3. Statistical Properties of Uncertainty Factors.**

| Variable | Mean | Standard Deviation | C.O.V. | Type of Distribution |
|----------|------|--------------------|--------|----------------------|
| $V_w$ | 1 076.05 x10$^6$ | 215.3 x10$^6$ | 0.25 | Normal |
| $H_w$ | 25 | 2.500 | 0.30 | Normal |
| C | 2 | 0.140 | 0.070 | Normal |
| L | 116.00 | 1,74 | 0.0121 | Normal |
| $H_{i+1}$ | 7.00 | 3.893 | 0.5561 | Normal |
| $t_*$ | 3600 | 360 | 0.10 | Normal |

**7.6 Excavation Conditions**
With decision taken to excavate a channel in the landslide, four stages were decided upon: condition A)
to reach 110 masl of elevation; condition B) to reach 100.00 masl in the excavations; condition C) to
perform the excavations until 96.00 masl; and finally, condition D) when excavations reach 92.00 masl.
With this value, if the water stored is released, the phenomenon is controlled in the Peñitas Dam, and
there is no more risk of failure downstream.
During the analysis, it is considered that the Peñitas spillway completely opens its gates at the beginning
of the landslide.
**7.7 Dam Overtopping Risk Analysis**
**7.7.1 Reliability**
The reliability of a system in civil engineering design (Tang, Ang, 1984) "is more realistic if measured in
terms of probability." The objective of a reliability analysis is to assure the event (X>Y), with X being
the supply capacity and Y being the demand capacity throughout the service life or some specified period
of the engineering system.



Traditionally, in a supply and demand problem, reliability has been expressed as a safety factor F=X/Y
or a safety margin M=X-Y, whereas the variables F, M, X, and Y are considered simple deterministic
variables.
If the supply or demand variables have a random nature, *F* and *M* become random variables as well.
Usually when an analysis is performed with stochastic variables, the results are expressed in terms of the
reliability index β that is defined as the probability of the supply capacity of the system exceeding the
demand capacity.
In the specific application to hydraulic works (overtopping of dams and spillways), the reliability index
β can be expressed as a function of the probability P of the margin of safety β = 1 - P*(M)*, and P*(M)* is
equal to the probability of the occurrence of the floods, X >*Y* if the discharge capacity of the spillway is
a deterministic quantity (Marengo, 2006).
The approximation presented herein considers that the floods produced by the overtopping of the natural
dam, as well as the reservoir levels, are estimated as stochastic random variables, and is applied the
Advanced First Order Second Moment method (AFOSM).
**7.7.2 Risk Analysis for the Peñitas Dam**
After a landslide is produced by any cause (overtopping, piping or sliding), a large flood happens against
Peñitas Dam, being overtopping the main risk if the spillway is not capable of evacuating the incoming
flood. The behavior function with the safety margin, is defined as:
$$M = H_R - H_{i+1} \tag{61}$$
where $H_R = (Z_{crown} - Z_{cv})$, with $Z_{crown}$ being the elevation of the dam crown (98.00 masl), $Z_{cv}$ being the dam
crest elevation of the spillway (elevation of 76.50 masl) and $H_{i+1} = \bar{H}_*$ being the highest water level
during the flood event which is given by (Eq. 52), evaluated with the consideration of the uncertainty
factors. In this case, the evaluation flood event does not correspond to the maximum hydrological flood
because it is produced by the dam break, and the events are independent one of each other.
Eq. (61) corresponds to the safety margin in the risk analysis and allows calculating the failure
probabilities of the system.
When landslide begins, the level of Peñitas water is 85.00 masl, which corresponds to the minimum of
operation as is shown in Fig. 7.




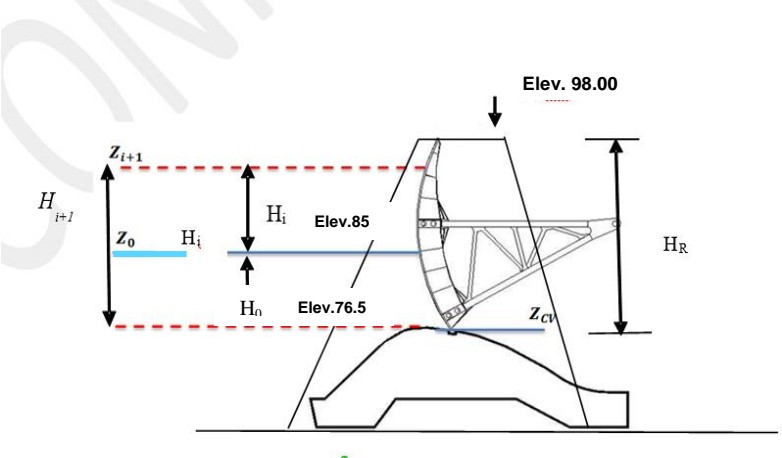

Where:

$$H_i = Z_{i+1} - Z_0$$
$$H_* = H_{i+1} = H_i + H_0 \qquad (62)$$


**8. Results and Discussions**
The risk analysis is made only with the Hagen, Costa, De Lorenzo and Mc Donald equations because they
offer the highest discharge values, considering six uncertainty factors, the AFOSM method was applied
to evaluate overtopping failure risk of the Peñitas Dam located downstream from a natural dam produced
by the Juan de Grijalva landslide. With the described methodology, it is possible to estimate the magnitude
of the flood peak, the return period, the probability of failure, the discharge flood in the Peñitas Dam, the
base time of the floods and the reliability index. Results are shown in Table 4 for the elevation of 110
masl.
**Table 4*. Results of ordinary risk with water elevation at 110 $masl$, Initial water***
***elevation $Z_0$= 85.00 masl.***






Hagen and Costa methods show a difference of 20% in discharge flows values, and failure probabilities differ by 326%. In addition, landslide flows with De Lorenzo and Mc Donald (a) are very similar and the probabilities of failure differ by 11.11 %, being greater than the De Lorenzo method.

The failure probability with the Hagen equation is very high (1.569 %), and the return period of 63.73 years is inadmissible for this kind of event. The evident decision was to continue excavating as the best action for reducing the failure probability of the system.

| \multicolumn{8}{l}{Upstream Volume $Vw=1076.9x10^6$ $m^3$; $\sigma_{vw}=269.22x10^6$ $m^3$ $Hw=25.00m$, $\sigma_{Hw}=7.500m$, $Z_0=85.00$ masl, $Hi=7.00m$, $\sigma_{Hi}=3.894m$, $t_{bl}=2.00hr$, $COV_{tbl}=0.10hr$.} |
|---|---|---|---|---|---|---|---|
| CODE | Eq. | $Q_c$ | $T_r$ (years) | $P_F$ | $Q_{vp}$ | $t_{bl}$ | $\beta$ |
| A-1 | Hagen | 121,322 | 63.731 | 0.01569 | 23,326 | 2.1430 | 2.1522 |
| A-2 | Costa (a) | 101,746 | 271.771 | 0.00368 | 23,456 | 2.1878 | 2.6801 |
| A-3 | De Lorenzo | 74,085 | 2 802.742 | 0.00036 | 23,630 | 2.2528 | 3.3843 |
| A-4 | Mc Donald (a) | 75,819 | 2 512.688 | 0.00040 | 23,626 | 2.2500 | 3.3542 |

Table 5 shows results for the 100.00 masl elevation.

**Table 5.** *Results of ordinary risk with water elevation at 100 masl. Initial water elevation $Z_0$= 85.00 masl.*

| \multicolumn{8}{l}{Upstream Volume $Vw=576.4x10^6$ $m^3$; $\sigma_{vw}=144.10x10^6$ $m^3$ $Hw=15.00m$, $\sigma_{Hw}=4.500m$, $Z_0=85.00$ masl, $Hi=7.00m$, $\sigma_{Hi}=3.894m$, $t_{bl}=2.00hr$, $COV_{tbl}=0.10hr$.} |
|---|---|---|---|---|---|---|---|
| CODE | Eq. | $Q_c$ | $T_r$ (years) | $P_F$ | $Q_{vp}$ | $t_{bl}$ | $\beta$ |
| B-1 | Hagen | 70,369 | 3 940.164 | 0.000254 | 23,652 | 2.2663 | 3.4768 |

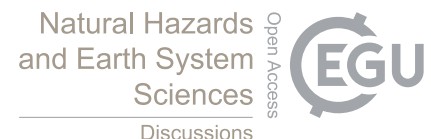

| | | | | | | | |
|---|---|---|---|---|---|---|---|
| B-2 | Costa (a) | 61,898 | 7 781.640 | 0.000129 | 23,632 | 2.2773 | 3.6552 |
| B-3 | Mc Donald (a) | 46,680 | 11 586.476 | 0.000086 | 23,300 | 2.1974 | 3.7562 |
| B-4 | De Lorenzo | 39,460 | 9 898.063 | 0.000101 | 23,135 | 2.0422 | 3.7164 |

Hagen method offers the highest value of failure probability (0.0254 %); the flow produced by a landslide is 70,369 $m^3/s$, which would produce the overtopping of the Peñitas Dam. The return period for the Hagen method is 3,940 years and 7,782 years for the Costa (a) method. Furthermore, the Mc Donald (a) method gives a value higher than 10,000 years, and the De Lorenzo method offers a return period of 9,898 years. The flows over the Peñitas spillway are similar and higher than 23, 135 $m^3/s$.

Excavating 507,680 m³ of earth and rock, reaching an elevation of 100.00 masl, significantly reduces the probability of failure, but still some of them reach greater values to usual return period (10,000 years).

Table 6 shows results for the 96.00 masl elevation.

**Table 6.** *Results of ordinary risk with water elevation at 96.00 masl. Initial water elevation $Z_0$= 85.00 masl.*

| Upstream Volume Vw=400.52x10⁶ m³; σvw=100.13x10⁶ m³ Hw=11.00m, σHw =3.30m, Z₀ =85.00 msnm, Hi=6.50m, σHi=3.894m, tbl=2.00 hr, COVtbl=0.10 hr. | | | | | | | |
|---|---|---|---|---|---|---|---|
| *CODE* | *Eq.* | *$Q_c$* | *$T_r$ (years)* | *$P_F$* | *$Q_{vp}$* | *$t_{bl}$* | *β* |





| C-1 | Hagen | 51,164 | 12,935.401 | 0.000077 | 23,456 | 2.1916 | 3.7836 |
| C-2 | Costa (a) | 46,195 | 11,522.813 | 0.000087 | 23,288 | 2.1220 | 3.7548 |
| C-3 | Mc Donald (a) | 35,410 | 12,604.971 | 0.000079 | 23,162 | 1.9019 | 3.7772 |
| C-4 | De Lorenzo | -- | -- | -- | -- | -- | -- |


The Hagen and Costa (a) methods exhibit the higher probabilities of failure but, are very similar. The
flows over the natural dam differ 0.76 %, and durations of the flows are quite similar.
The return periods already are little higher than 10,000 years, and it is possible to say that with this action
the emergency was attended; however, the decision to excavate an additional stage until reach   an
elevation of 92.00 masl was taken (Fig. 5).
Excavating an additional volume of 340,591 m$^3$ of earth and rock to reach an upstream elevation of 96.00
masl reduced failure probabilities significantly to values less than $1.00 \times 10^{-4}$ for all methods.
It is important mention that on December 2007, there was no more intense rainfall in the zone, the decision
was taken to continue with the excavation force reaching an additional retired volume of 369,290 m$^3$ (with
a total excavation volume of 1,217,561 m$^3$). In addition, the final 92.00 masl elevation was reached,
opening the excavated channel to flow on December 18th of that year.
The analysis showed herein establishes that the system with an upstream elevation of 110 masl and a
water volume of 1076.90 $\times 10^6$ m$^3$ presents inadmissible conditions of risk with values of risk as 1.569%,
which is extremely high for the analyzed situation. The methodology presented permits the engineer to
take decisions of progressive excavations to solve this extremely delicate situation.

**8.1 Sensitivity analysis**
**8.1.2 Duration of flood variation**





This variable is the most important because during analysis is inherent to the phenomena and cannot be
controlled from the human point of view; however, it has a significant importance during event
occurrence.
The sensitivity analysis is done considering several possible durations of flood landslide (1.00 hr, 1.50
hr, 2.50 hr and 3.00 hr) and initial upstream water elevations for 110.00 masl, as well as the volume
$=1,076.9 \times 10^6$ m$^3$; the results are shown in Table 7.

**Table 7. Initial 110.00 masl elevation, Vw=1076.9x10$^6$ m$^3$; Hw=25.00m, t$_{bl}$=1.00 hr, t$_{bl}$=1.50 hr,**
**t$_{bl}$=2.00 hr,  t$_{bl}$=3.00 hr.**


Comparison permits observing that failure probabilities obtained with the Hagen method increase
significantly when durations change from 1.00 hr to 3.00 hr, with a change of 10,263.702%. In addition,
there is a difference of 2,967% with the Costa (a) criterion. For durations of 2.5 hr and 3.00 hr, there were

| | t$_{bl}$= 1.00hr | | t$_{bl}$= 1.50hr | | t$_{bl}$= 2.00hr | | t$_{bl}$= 2.50hr | | t$_{bl}$= 3.00 hr | |
|---|---|---|---|---|---|---|---|---|---|---|
| Ec. | P$_F$ | t$_{bl}$ | P$_F$ | t$_{bl}$ | P$_F$ | t$_{bl}$ | P$_F$ | t$_{bl}$ | P$_F$ | t$_{bl}$ |
| Hagen | 0.001394 | 1.0511 | 0.004787 | 1.6017 | 0.01569 | 2.1430 | 0.04979 | 2.6520 | 0.14447 | 3.1198 |
| Costa (a) | 0.000681 | 1.0513 | 0.001625 | 1.6145 | 0.00368 | 2.1878 | 0.00847 | 2.7487 | 0.02089 | 3.2757 |
| Mc Donald (a) | 0.000254 | 1.0458 | 0.000355 | 1.6195 | 0.00036 | 2.2528 | -- | -- | -- | -- |
| De Lorenzo | 0.000270 | 1.0464 | 0.000385 | 1.6200 | 0.000398 | 2.2500 | -- | -- | -- | -- |

no results obtained for the Mc Donald (a) and De Lorenzo equations.



Such a large obtained variation is transcendent for the risk analysis and decisions for attending a case like the one discussed here. It is emphatically recommended that scenarios analysis be made with different flood durations of the hydrograph studied, it is remarkable that the estimation must be carefully studied in real cases and with physical models to increase knowledge regarding the explanation of this kind of phenomena. One main suggestion is trying to do an intensive program, perhaps with small prototype models and make an experimentation program to gain better knowledge of the behavior of this variable. In any event, the results of failure probabilities obtained with this risk model analysis let us conclude that in similar cases it is necessary to do all possible efforts to undertake large excavations to reduce the floods produced by landslides.

### 8.1.3 Hydraulic Head Variation in the Reservoir

Observing director cosines of the AFOSM method used, it is observed that the hydraulic head in downstream reservoir has more significance than other variables; taking this fact into account, a sensitivity analysis was done with different values for standard deviation and coefficients of variation conserving a mean value of $h_i$= 7.00 m, $\sigma_H = 4.20\ m\ (C.O.V. = 0.60)$, $\sigma_H = 4.55m\ (C.O.V. = 0.65)$, and $\sigma_H = 4.90\ m\ (C.O.V. = 0.70)$. The initial elevation analyzed was 110 masl, and the water volume was 1,076.9 x $10^6$ m³. Results are shown in Table 8.

**Table 8. Initial elevation of 110.00 masl, with h=7.00m and $t_{bl}$=2.00h, Vw=1076.9x$10^6$ m³; $\sigma_{vw}$=269.22x$10^6$ m³, Hw=25.00m, $\sigma_{Hw}$ =7.500m, C.O.V.=0.60, COV=0.65, COV=0.70.**

| | H=7.00m, $\sigma_H = 4.20$m COV=0.60 | H=7.00 m, $\sigma_H = 4.55$m COV=0.65 | H=7.00, $\sigma_H = 4.90$m COV=0.70 |
|---|---|---|---|
| | | | |




| Eq. | Qc | Tr (years) | PF | Qc | Tr (years) | PF | Qc | Tr (years) | PF |
|---|---|---|---|---|---|---|---|---|---|
| Hagen | 121,513 | 42.95 | 0.02329 | 121,636 | 30.034 | 0.03329 | 121,733 | 22.505 | 0.04443 |
| Costa (a) | 101,877 | 148.89 | 0.006717 | 101,989 | 86.471 | 0.011564 | 102,074 | 56.176 | 0.017080 |
| Mc Donald (a) | 74,166 | 1,045.38 | 0.000957 | 74,232 | 440.301 | 0.002271 | 74,281 | 225.179 | 0.00444 |
| De Lorenzo | 75,925 | 948.94 | 0.001054 | 75,689 | 404.572 | 0.002471 | 75,797 | 208.963 | 0.00479 |


Failure probabilities increase is notorious; Hagen method goes from the 1.569 % (C.O.V.=0.5561)
initially obtained to 4.44 % with a C.O.V.=0.70 (a difference of 183.17%). Furthermore, the increase for
the Costa (a) method is 364% for a C.O.V.=0.70.
With Hagen's equation, the return period diminishes from 64 years to 23 years with a C.O.V.=0.70 and
from 278 years to 56 years for the Costa (a) method.
The obtained results permit conclude that is necessary to do a strong effort to study this variable in a better
way, for example, with bathymetry studies and satellite tools. Obviously, it is recommended to study the
uncertainties with different methodologies, for example, the Rosenblueth point estimation method (1985),
Harr's point estimation method (1987), the Monte Carlo simulation (Chang, 1994), and the Latin
hypercube sampling (Jan-Tai Kuo, 2007).
**9. Application to similar situations**
Engineers cannot have intervention in diverse types of variables; however, it is possible to increase a dam
crest if a flood of a natural dam is produced upstream. In addition, when there is the certainty that an
upstream dam can fail, it is possible to build an upper elevation on downstream dams with sandbags, or
if there is time enough, to use machinery with proper material for doing an upper elevation.
Analysis performed is shown in Table 9, failure probabilities, return periods and final duration floods for
Peñitas Dam, if the crown should reach 98.50 masl (50 cm over Peñitas crown).






**Table 9. Comparison between the dam crown at an elevation of 98.00 masl and at 98.50 masl in the Peñitas Dam**

| Eq. | Elev. 98.00 masl H=7.00m, $\sigma_H = 3.8945$ $t_{bl}$=2.00 hr | | | Elev. 98.50 masl H=7.00m, $\sigma_H = 3.8945$m $t_{bl}$=2.00 hr | | |
|---|---|---|---|---|---|---|
| | $P_F$ | Tr (years) | $t_{bl}$ | $P_F$ | Tr (years) | $t_{bl}$ |
| Hagen | 0.01569 | 63.731 | 2.1430 | 0.009679 | 103.312 | 2.1588 |
| Costa (a) | 0.00368 | 271.771 | 2.1878 | 0.002079 | 480.998 | 2.2054 |


Results with Hagen's method showed that failure probabilities reduce to 62.10 %, and the return period went from 63.73 years to 103.312 years. With the Costa (a) method, the failure probabilities were reduced to 77.01 %, and the return periods went from 271.77 years to 480.99 years.

The results with the Mc Donald (a) and De Lorenzo methods showed failure probability reductions of 108 %, and the return period is almost twice the original value.

The Increase of the dam crown is very important. In addition, achieving this simple action is relatively easy, and it allows increasing safety in the downstream dam.

580

581

**CONCLUSIONS**

Dam break situations are one of the most devastating phenomena for any society. In this study, a mathematically explicit evaluation of the upstream water level of a dam is proposed that receives a flood produced by a natural landslide dam break, and how doing risk analysis of this complex phenomena. This



paper demonstrates the procedure for evaluating the risk to the Peñitas Dam in Tabasco, Mexico, when a natural dam produced by a landslide can fail. Each uncertainty analysis has its own hypotheses, limitations, advantages and disadvantages. The AFOSM method used here can yield accurate and logical estimations, including random variables; however, as the nonlinearity or factor uncertainty level increases, the accuracy of certain situations deteriorates.

It was demonstrated during risk analysis that there are significant variables that intervene in the risk analysis, and they have a significant weight over the obtained results. Therefore, duration flood and the upstream water elevation of the reservoir were identified as the most significant during the sensitive analysis; in addition, it was found that excavating the natural dam at different stages was the best solution for the analyzed situation.

The upstream solution, reaching a 92.00 $masl$ elevation, solved the emergency returning to operation conditions and nevertheless under very risky conditions, it was possible to solve the situation for the downstream population.

1.	It is desirable that besides calculating with methodology developed like is shown in this document, other methodologies with fast applications can be proposed and analyzed so that engineers can confront similar situations and can rapidly act to solve them.

2.	May be very interesting to use other methodologies that can evaluate uncertainties, such as Rosenblueth´s point estimation method, Harr's point estimation method, the Monte Carlo simulation, and the Latin hypercube sampling, and in all cases that can systematize risk evaluation and that can improve knowledge about uncertainties that are present in these cases.

3.	Without a doubt, these kinds of risk analysis will let engineers make better decisions and improve solving in a better way these kinds of situations that surely will continue happening in the world future.

4.	When a similar situation is presented, recommendation of upper elevation of the downstream dam should be applied immediately to gain safety.

5.	Risk analysis methodology like herein presented permits performing orderly and consistent decisions necessary for solving this kind of events.



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
