# Peer review of "Case Study: Risk Analysis by Overtopping During an Upstream Landslide in Peñitas Dam, Mexico"

_Natural Hazards and Earth System Sciences, 2019_

## Referee Comment (RC1) · Anonymous Referee #1 · 23 Aug 2019

The presented manuscript deals with the case study of Penitas dam, where a landslide blocked the Grijalva river and lead to severe problems, not only for the adjacent hydroplants but for general safety of downstream inhabitants. The work is separated in two sections: the derivation of an explicit expression for maximum flood head calculation and the case study analysis where different peak flow regimes taken from literature are considered.

Generally speaking, the manuscript is extremely difficult to follow. First and foremost, the linguistic quality is so low, that it impedes a fluent reading an often obstructs the clarity of the made statements. Furthermore, the two parts stand almost detached

from each other (or it just was not clear to my confused eyes) such that the work itself is hard to grasp and to deduce the relevant and important novel findings. Moreover, the presentation quality is such, that paragraphs and entire sections are missing, legends of tables are floating around and typos are ubiquitous.

As NHESS is kind of a general audience publication, generally, content like the proposed merits publication, but only if substantially tuned to a broader readership. The numerical derivation of the maximum flow head should be presented in a more specific journal, where the review process can also judge more accurately whether the derivation is novel and scientifically sound. To be honest, not every equation was double checked within this review process, but I sincerely believe that a paper with 52 equations is not suitable for publication in NHESS.

While many topics are touched as well as in the numerical part as well as in the case study part, the presented work is poorly linked to existing/similar work.

As can be found in the technical corrections (see attached file), the list is exorbitantly long. It kind of felt as I performed the first proof reading. With all due respect to the authors, this is not an acceptable manuscript quality to be submitted to any journal. A proof reading at least from a native speaker is required, such that a referee can deal with the scientific content rather then being language corrector.

As NHESS stand for generally high quality publications, I have no alternative but to recommend that the manuscript must be rejected.

If the authors decide to re-submit the work I would propose the following: Split the work. Send the numerical part to a dedicated hydraulic journal and let it be reviewed there. The case study part fits to NHESS, but needs major revisions with respect to other publications and events. Generally, spend more time and effort in preparing a readable manuscript.

Please also note the supplement to this comment:

[Figure]

https://www.nat-hazards-earth-syst-sci-discuss.net/nhess-2019-191/nhess-2019-191-RC1-supplement.pdf

[Figure]

**Supplement:**

**Technical corrections:**

**Abstract:**

l8: for the analysis six statistical variables and their uncertainties are used: …

l9: clarify the sentence

l11: Afterwards → Subsequently,

l12: for solve the problem → in order to solve the problem/as countermeasure.

l13: of duration of dam break →damn break duration/volume/ ?

l13/14: sentence is unclear: A possible lower limit for the damn elevation is calculated?

l14: highly recommending on-site measurements/damn elevation monitoring ??

→ rephrase and make it more concise and clear.

l15: A methodology based on what? → end the abstract with a powerful statement and not with a generic one.

1. **Introduction**

l18 higher flood compared to what? Highest ever? Higher than normal, higher by how much?

L19: a landslide took place … .

l20 made a dam → created a dam

l21 The landslide thickness amounted to 80(space) m covering an area of 800 m length and 300 m width.

l22 fell over? → covered the river bed, covered the slope areas?

l24 in view that was not possible → because the water drainage of the last reservoir was impossible (due to debris ?)

l25 completing → completely

l26 The imminent failure of the naturally created dam posed a severe threat ?

would it impose hydrological consequences? Would it take out the Hydroelectric Power House ? What are the severe consequences? → you state it in l27-31, so maybe just delete this sentence here.

l31 amounting to 77% of hydroelectrically used water runoff in Mexico? Or is ist 77% of the total water flow in Mexico? Clarify for reader unfamiliar with local geography.

l32-35: If you want a detailed introduction of Grijalva river, place it at l19, where you introduce the water catchment system.

2. **Landslide**

l37 a landslide happened

l38-39 units are bold, keep it uniform and not bold text

l39. Malpaso dam. Initial mass movement was the detachment of a rock block of 1300 m length and 75 thickness consisting of limestone and sandstone rocks covering the river slope.

l43 produced a natural dam ow 80 m height, 800 m length and 300 width.

→ indicate the lenghtscales in Fig. 1b and reference to it in the text.

3. **Where is Paragraph 3.?**

4. **Geological framework**

l46: The landlside …

l45: … base, a stratigraphic formation prone to act as a lubricant if subjected to heavy rainfalls as they occured during the end of October until beginning of November 2007.

l51 8° to 11°

l50ff: (1) was it the lubricated layer? (3) Why highy local stress/pressure relief in the beginning? (5) high water table (water level) ? clarify?

**4.1 Geological Model of the failure mechanism**

If there is no Section 4.2 it does not make much sense to have a Sec. 4.1. Streamline.

l58 according to the scale Carson and Kirkby

l60 initiated a slow movement

l61 occasioned → yielded

l62 diminished shear resistance

l64 lower portion, lower section? Unclear formulation of the entire sentence.

The lower section of the sliding rock mass turned into a heavy viscous mass consisting of debris and rock boulders producing a 50 m high debris wave (?). This wave buried a small village killing 25 people and completely obstructing the river path. ???

5. **Basis of the Study**

l102 used for water supply

l103 the various safety level (?), limited to what extent?

l108 clear understanding of embankment failure processes *or* a accurate prediction of embankment failure processes

l117-121 citation should read as followingly: *"failure algorithms of low levels of complexity are still needed when detailed simulations are not required or are not possible to apply easily or conveniently. For these reasons, a simple empirical model that considers a breach to form in a presupposed way, usually growing in the shape of a trapezoid is applied often in practice"*

l126 as can be their implications on measures to minimize flood hazards.

L128 predicted dam outflow hydrographs, peak flow levels and flow rates at downstream location of interest

6. **Approach to the Problem**

L131 the overtopping analysis is performed in the following sequence:

1) the flood routing over Penitas dam is defined under the development of an explicit analytical description leading to estimations of water masses flowing through the spillway (?)

2) empirical methods are deployed for peak flood estimation

3) behaviour function (?) → flow regime description is obtained used for risk assessment (of what)

→ please itemize correctly with correct structure: noun (with adverbs/adjective), verb and then the rest of the sentence

l135 the methodology is applied to different excavation conditions → this is not belonging to the initial analysis sequence but is an application of it.

l136 5) is also an application → Subsequently the methodology is applied as well for the upper elevation (what do you mean with that: upstream dam, Malpaso dam failure scenario? Clarify.

**6.1 Flood routing**

l140 Eq (1): $Q_I$ and $Q_f$ are incoming fluxes, $Q_S$ outgoing flux. Correct? Indication with the correspondent sign would clarify the situation.

l144 t is the the time, dt is the time derivative. Maybe just introduce dS/dt as the temporal change of stored water.

**6.2 Storage Capacity Curve**

l148 Subscript of $S_0$

l149 $S_F$ is the storage corresponding to elevation $Z_F$

l152 Taking the time derivative of Eq 2 yields

**6.3. Hydrograph produced by the landslide**

l162 why is that not the case for a different shape of hydrograph?

**6.4 Spillway discharge …**

l169 eq (6). Why is the spillway law like that, why is it $H^{3/2}$. For a general public reader as NHESS is aiming for, explain in more detail.

**6.5 Flood routing reviewed**

l186 here $CLH^{3/2}$ has a minus sign, so QS in equation (1) should have one too.

l188 … of the hydraulic head of the spillway H.

**6.6 Flood Routing Discretization**

l194 $F_D(H_m, H_{j+1}; \Delta t …) \equiv \alpha$ […]

l207: if t0 = 0, Eq. (19) yields:

l212 the equation of differences…

l213 that if it is possible to build a twice differentiable continuum function around it is fair to assume (?)

l216 I don't think "differences equation" the right term. Please check Numerical Method books for the correct English word.

l218 Please explain for reader not that familiar with numerical methods. We can build lots of stuff with a cubic spline…

l220 please specify the page of your reference. An entire book as reference is not so useful to check the made statement. Could you explain the reasoning behind the

truncated error as NHESS is not a numerical method paper per se.

l221 is not linear

l223 a similar strategy as proposed by

l242 the order of magnitude of the truncated error

l252 The truncated error is given by Eq (27):

l264 what do you mean with time design flood? Which two outflow must happen at the same time?

**6.7 Oridnary Risk Case**

l287-321 To be honest, I did not check every single equation here as I thinkt such a detailed presentation is not useful in a NHESS paper.

l321: Give in plain words the meaning and the leading influences for the maximum head equation, such that the interested reader can grasp the essence of it without spending dissecting the equation.

**7. Case Study**

**7.1 Water Level Upstream Elevations of Landslide**

l325 are as proposed in Fig. 6.

l340-345 clarify the formulation: what was excavated?

l345 the spillway has the capacity of discharge of which level with no risk? Elev 92?

**7.2 Empirical Peak flow Estimations of Dam Failure**

l348 dam break

l352 in the table: failure dams → dam failures

l359 equation, it can be seen that

l363 why are the largest values chosen? For a worst case scenario? Why are the lower ones not accurate/too optimistic? Can you explain?

**7.2 Landslide Duration**

l367/368 very reduced → heavily reduced/suppressed/diminished

l369 the water height/level was similar as the one presented in our case study/in the Penitas dam failure.

Generally: what is the link of the literature findings to the presented Penitas case? In a chapter "Landslide Duration" within a paper of a specific case study, an estimation /back-calculation/comparison with different case studies is preferable.

**7.4 Analysis of Statistical Variables**

l382 maybe put the standard deviation again into context with the actual upstream reservoir capacity (in numbers) such that the reader does not have to flip back and forth and recheck the numbers.

l380-394/Table 3 Again: As NHESS is not a journal solely devoted to hydraulic processes,

where do the standard deviations originate from, are the COV coefficients accepted best practice in this field. Please elaborate for a broader readership.

**7.5 Analysis Considerations**

→ an entire subsection just for one sentence is a bit overkill

**7.7 Dam Overtopping Risk analysis**

l430 and the Advanced Firs… (AFOSM) is applied.

**7.7. Reliability**

l431 → show how the AFOSM method has been used for dam overtopping, **AFSOM cite relevant work 8** Ganji, A. & Jowkarshorijeh, L. Stoch Environ Res Risk Assess (2012) 26: 33. https://doi.org/10.1007/s00477-011-0517-1

l441 independent of each other

8. **Results and Discussion**

l463 in discharge flow values

Table 4: make sure the legend is close by, label the variable names for easier readability

l466-507 clarify the nomenclature: 70k m/s → 70'000 $m^3$/s, ordinary decimal point 0.0254%) etc → readability increases if only decimal points are on the lower line IMHO.

l496 the emergency state was amended, until the reach of an/reaching an elevation of

l500 it is important to mention, zone → area

l501 force? Retired?

l503 to release the routing through the excavated channel on Dec…

**8.1 Sensitivity analysis**

→ where is 8.1.1?

l511 repeat the variable. Only the title does not count as introducing the variable

l511-513 rephrase, unclear.

l516 volume V = $1.0769 \times 10^9$ $m^3$

Ll518 legend of table should be next to the table

l527 emphatically? Delete that.; scenario analysis

l531 is there no literature out there trying to do that?

L536 what are director cosines? What are you observing in them? How are they telling us, that the downstream reservoir is more significant?

L557 permit to conclude

9. **Application t similar situations**

l563 Engineering interventions are limited to certain variables.

l564 when it is certain that

l567 Table 9 shows failure probabilities, return periods and …

l578 the increase

Conclusions

l583 for downstream population.

l585 the risk analysis of such a complex phenomenon.

l587-590 generic comment. What are the advantages and uncertainties in this presented study.

l595 compare to other solutions possible?

L599-611 Do not itemize. The conclusion section should be written in a coherent, concise way.

l602-605 Might be interesting, however, the general reader of NHESS needs short comparison of the methods. To generic statement as a NHESS conclusion.

---

## Short Comment (SC1) · 30 Aug 2019

**Flood Routing**

The reservoir routing follows continuity equation:

$$\frac{dS}{dt} = Q_l + Q_f + Q_s \tag{1}$$

where S is the storage in the reservoir of the Peñitas Dam, $Q_l$ is the flow generated by the landslide, $Q_f$, is the flow of tributaries rivers to the site of Peñitas, $Q_s$ is the flow extracted from the Peñitas spillway, and $t$ is the analysis time.

**6.2 Storage Capacity Curve**

Storage capacity elevation curve for the reservoir may be expressed as:

$$\frac{S-S_o}{S_F-S_o} = \left(\frac{Z-Z_o}{Z_F-Z_o}\right)^\alpha \tag{2}$$

where $Z$ is the elevation of the free water surface in the reservoir, $S_o$ is the storage corresponding to $Z_o$ elevation, which will be considered as a conservation level, $S_F$ is storage corresponding $Z_F$ elevation, which can be interpreted as the maximum level that can be reached when Eq. (1) is solved, $\alpha > 1$ is a regression constant.

From Eq. (2),

$$\frac{dS}{dt} = \alpha \frac{S_F-S_o}{Z_F-Z_o}\left(\frac{Z-Z_o}{Z_F-Z_o}\right)^{\alpha-1}\frac{dZ}{dt} = \alpha \frac{S_F-S_o}{Z_F-Z_o}\left(\frac{Z-Z_o}{Z_F-Z_o}\right)^{\alpha-1}\frac{dH}{dt} \tag{3}$$

where:

$$H = Z - Z_{cv} \tag{4}$$

is the spillway crest head and $Z_{cv}$ is crest elevation.

**6.3 Hydrograph produced by the landslide**

According to Fig. 3, the flow produced by the landslide can be written as

$$Q_l(t) = \begin{cases} 0, t \in (-\infty, 0) \\ Q_{pl}\left(1 - \frac{t}{t_{bl}}\right), t \in (0, t_{bl}) \\ 0, t \in (t_{bl}, \infty) \end{cases} \tag{5}$$

where $Q_{pl}$ is the peak flood and $t_{bl}$ is the base time of the hydrograph. It must be noted that the triangular form of the hydrograph permits an increase in the volume if it is necessary.

[Figure]

**Figure 3 Discharge law of the hydrograph**

**6.4 Spillway discharge for the Peñitas Dam**

The spillway discharge is shown in Fig. 4 and is given by

$$Q_s = \begin{cases} 0, & H < H_o \\ CLH^{\frac{3}{2}}, & H \geq H_o \end{cases} \tag{6}$$

where

$$H_o = Z_o - Z_{cv} \tag{7}$$

$C$ is the discharge coefficient, and L is the spillway length.

Note that if

$$Q_s < Q_l + Q_f, \quad t \in \left(0, t_{pf}\right) \tag{8}$$

then Eq. (6) may be written as

$$Q_s = \begin{cases} 0, & t < 0 \\ CLH^{\frac{3}{2}}, & t \geq 0 \end{cases} \tag{9}$$

In fact,

$$Q_{s,o} \equiv CLH_o^{3/2} \tag{10}$$

is the discharge in the spillway when $t=0$, as is shown in Fig.4.

[Figure]

**Fig. 4. Discharge Law of the Spillway**

**6.5 Flood routing reviewed**

By substituting Eqs. (3) and (10) in Eq. (1),

$$F_c(H) = \alpha \frac{S_F - S_o}{Z_F - Z_o} \left(\frac{Z - Z_o}{Z_F - Z_o}\right)^{\alpha - 1} \frac{dH}{dt} - \left[Q_l(t) + Q_f(t) - CLH^{\frac{3}{2}}\right] = 0, t > 0 \qquad (11)$$

where $Q_l(t)$ y $Q_f(t)$ are given by Eqs. (5) and (6), and $F_c(\cdot)$ is a differential operator that acts over the hydraulic head of the spillway, $H$.

**6.6 Flood Routing Discretization**

Eq. (13) has no analytical solution for an arbitrary value of α. Thus, a discretization solution based on the trapezoidal rule is done:

$$F_D = (H_j, H_{j+1}; \Delta t_{j+1/2}) \equiv \alpha \frac{S_F - S_o}{Z_F - Z_o} \left[\frac{1}{2}\left(\frac{Z_c + H_j - Z_o}{Z_F - Z_o}\right)^{\alpha - 1} + \frac{1}{2}\left(\frac{Z_c + H_{j+1} - Z_o}{Z_F - Z_o}\right)^{\alpha - 1}\right] \frac{H_{j+1} - H_j}{\Delta t_{j+\frac{1}{2}}} - $$

$$\left[\frac{Q_{l,j} + Q_{l,j+1}}{2} + \frac{Q_{f,j} + Q_{f,j+1}}{2} - \frac{CL}{2}\left(H_j^{3/2} + H_{j+1}^{3/2}\right)\right] = 0; \ j = 0, 1, \dots. \qquad (12)$$

where

$$H_j \approx H(t_j) \qquad (13)$$

$$H_{j+1} \approx H(t_{j+1}) \qquad (14)$$

Both are discrete approximations of the head values over the spillway crest in time $t_j$ and $t_{j+1}$. Thus,

$$Q_{l,j} = Q_l(t_j) \tag{15}$$

$$Q_{l,j+1} = Q_l(t_{j+1}) \tag{16}$$

$$Q_{f,j} = Q_f(t_j) \tag{17}$$

$$Q_{f,j+1} = Q_f(t_{j+1}) \tag{18}$$

[revised manuscript text omitted]

---

## Author Comment (AC1) · 31 Aug 2019

**Flood Routing**

The reservoir routing follows continuity equation:

$$\frac{dS}{dt} = Q_l + Q_f - Q_s \tag{1}$$

[revised manuscript text omitted]

$f(x)$ can be written as

$$f(x+\Delta x) = f(x) + f'(x)\Delta x + \frac{1}{2} f''(\xi)\Delta x^2, \quad x < \xi < x + \Delta x, \tag{22}$$

100 where the residue has been written in a Lagrangian form.

101 By identifying $x$ with $H_j$ and $f(x)$ with $\left(\frac{Z_c+H_j-Z_o}{Z_F-Z_o}\right)^{\alpha-1}$, as well as $\Delta x$ with $H_{j+1}-H_j$, the

102 Taylor theorem (22) can be written as

103 $\left(\dfrac{Z_c+H_{j+1}-Z_o}{Z_F-Z_o}\right)^{\alpha-1} = \left(\dfrac{Z_c+H_j-Z_o}{Z_F-Z_o}\right)^{\alpha-1} +$

104 $(\alpha-1)\dfrac{(Z_c+H_j-Z_o)^{\alpha-2}}{(Z_F-Z_o)^{\alpha-1}}\left(H_{j+1}-H_j\right) + \dfrac{(\alpha-1)(\alpha-2)}{2}\dfrac{(Z_c+H_{j+\beta}-Z_o)^{\alpha-3}}{(Z_F-Z_o)^{\alpha-1}}\left(H_{j+1}-H_j\right)^2;$   (23)
$$0 < \beta < 1$$

105 Now identifying $x$ with $H_j$, $f(x)$ with $H_j^{3/2}$ and $\Delta x$ with $H_{j+1}-H_j$ for known $H_j$, it is possible

106 again to apply Taylor's theorem (22) as

107 $$H_{j+1}^{3/2} = H_j^{3/2} + \frac{3}{2}H_j^{1/2}\left(H_{j+1}-H_j\right) + \frac{3}{8}H_{j+1}^{-\frac{1}{2}}(H_{j+1}-H_j)^2; 0 < \gamma < 1 \qquad (24)$$

108 Obviously

109 $$H_{j+1} - H_j = O(\Delta t_{j+\frac{1}{2}}) \qquad\qquad (25)$$

110 By substituting Eqs. (23) and (24) in Eq. (22) and considering the definition of differences

111 $F_D$ given in Eq. (12), then:

112 $F_D = \left(H_j, H_{j+1}; \Delta t_{j+\frac{1}{2}}\right) \equiv \alpha\,\dfrac{S_F-S_o}{Z_F-Z_o}\left[\left(\dfrac{Z_c+H_j-Z_o}{Z_F-Z_o}\right)^{\alpha-1}\right]\dfrac{H_{j+1}-H_j}{\Delta t_{j+\frac{1}{2}}} - \left[\dfrac{Q_{l,j}+Q_{l,j+1}}{2} + \dfrac{Q_{f,j}+Q_{f,j+1}}{2} - \right.$

113 $\left. \dfrac{CL}{2}H_j^{\frac{3}{2}} - \dfrac{3}{4}CLH_j^{\frac{1}{2}}\left(H_{j+1}-H_j\right)\
[revised manuscript text omitted]

---

## Author Comment (AC2) · 1 Sep 2019

Mexico City 31 August 2019

Case Study: Risk Analysis by Overtopping During an Upstream Landslide in Peñitas Dam, Mexico Humberto J.F. Marengo1, Alvaro A. Aldama2

Dear Elena:

I am sending to you the following: 1. A short comparison of the methods such a general reader of NHESS can understand the language and the methodology stablished and be possible to generate a generic statement NHESS conclusion.

2. With regard to some technical observations made, I am sending a copy of 375 page of the Probability Concepts in Engineering Planning and Design of Ang & Tang (Vol.II, 1984) where is shown the concept of spillway discharge, that is common in hydraulic and civil engineering. Note: The copy is included at the end of the file.

Best regards.

Humberto Marengo

Please also note the supplement to this comment:
https://www.nat-hazards-earth-syst-sci-discuss.net/nhess-2019-191/nhess-2019-191-AC2-supplement.pdf

―――――――――――――――――――――

**Supplement:**

Comparison of Risk Methods.

Reviewer #1 wrote:

A brief comparison of reliability analysis methods must include also evaluation of uncertainty methods that are used for evaluation of a specific problem.

Answer could be:

Reliability analysis methods.

The risk associated with the potential failure of a hydraulic engineering system is the result of the combined effects of inherent randomness of external loads and various uncertainties involved in the analysis, design, construction, and operational procedures. Hence, to evaluate the probability that a hydraulic engineering system would function as design requires performing uncertainty and reliability analysis.

The reliability $P_S$, is defined as the probability of safety (or non-failure) in which the resistance of the structure exceeds the load, that is,

$$P_S = P(L \leq R) \quad (1)$$

Conversely, the failure probability, $P_F$ can be computed as:

$$P_F = P(L > R) = 1 - P_S \quad (2)$$

Accordingly, the reliability is a function of random variables:

$$P_S = P[g(X_{L)} \leq h(X_R)] \quad (4)$$

The methods for evaluating this equation are:

| Method | Observations |
|---|---|
| Direct Integration | The method requires PDFs be mathematical known or derived; very hard in real problems |
| | |
| Mean-Value First-Order Second-Moment (MFOSM) | Behavior function is expanded in a Taylor series at a selected point and evaluated with the first two statistical moments of random variables. Usually mean values of random variables |
| | |
| Advanced First-Order second-Moment (AFOSM) Method | The expansion of behavior function is made on failure surface. The point on this failure surface associated with the lowest reliability is the one having the shortest distance in the standardized space to the point where the means of the random variables are located. (Design point). This are named direct cosines to the expansion point. This method is applied to correlated and to no correlated normal random variables |
| | |
| Monte Carlo Simulation | This is a general method tp estimate statistical properties of a random variable that is related to a number of random variables which may o may not be correlated. The generated parameter values are generated according to their distributional properties. The major disadvantage is its computational intensiveness. There are some variations of this method, like Latin hypercubic sampling, importance sampling and the reduced space approach. |
| | |

The techniques for uncertainty analysis are:

| Analytical Techniques | Observations |
|---|---|
| Fourier and Exponential Transforms | Are particularly useful when random variables are independent and linearly related. The convolution property of the Fourier transform can be applied to derive the characteristic function of the resulting random variable. |
| | |
| Mellin Transform | When the random variables in a function are independent and nonnegative and the functions g(X) has a multiplicative form. |
| | |
| First-order Variance Estimation (FOVE) Method | The method approximates a model involving random variables by the Taylor series expansion. |
| | |
| Rosenblueth's Probabilistic Point Estimation (PE) Method | Is a computationally straight forward technique for uncertainty analysis. It can be used to estimate statistical moments of any order of a model output involving several random variables which are either correlated or uncorrelated. |
| | |
| Harr's Probabilistic Point Estimation (PE) Method | To avoid the computationally intensive nature of Rosenblueth's Method, when the number of random variables is large, Harr, proposed an alternative probabilistic method which reduces the required evaluations from $2^N$ to 2N and greatly enhances the applicability of the PE method of practical problems. |
| | |

$$\times \log\left[\frac{(3.72 - 0.0253\beta) + (0.5 + 0.0544\beta)}{3.72 - 0.0253\beta}\right] = 0$$

By trial-and-error: $\beta = 1.27$

[Figure]

| 2 | | | | | |
|---|---|---|---|---|---|
| $N$ | 1.037 | $-0.241$ | $-0.323$ | $1.00 + 0.0323\beta$ | $8\beta$ |
| $C_c$ | 0.487 | $-0.508$ | $-0.681$ | $0.396 + 0.0674\beta$ | $2\beta$ |
| $e_o$ | 1.137 | 0.209 | 0.280 | $1.19 - 0.05\beta$ | $6$ |
| $H$ | 169.60 | $-0.126$ | $-0.166$ | $168.00 + 1.394\beta$ | $3$ |
| $p_o$ | 3.688 | 0.117 | 0.157 | $3.72 - 0.0292\beta$ | $8\beta$ |
| $\Delta p$ | 0.569 | $-0.409$ | $-0.548$ | $0.5 + 0.0548\beta$ | |

Failure equation:

$$2.5 - (1 + 0.0323\beta)\frac{0.396 + 0.0674\beta}{1 + (1.19 - 0.05\beta)}(168 + 1.394\beta)$$

$$\times \log\left[\frac{(3.72 - 0.0292\beta) + (0.5 + 0.0548\beta)}{3.72 - 0.0292\beta}\right] = 0$$

By trial-and-error: $\beta = 1.27$

The failure point, therefore, is

$$n^* = 1.041, c_c^* = 0.482, e_o^* = 1.127, h^* = 169.8, p_o^* = 3.683,$$

$$\Delta p^* = 0.570$$

The probability of excessive settlement then is

$$p_F = 1 - \Phi(1.27) = 0.102$$

**EXAMPLE 6.12 (Spillway Capacity)**

Inadequate spillway capacity to carry the inflow water during an extreme flood is a major cause of dam failure. For a spillway with an uncontrolled overflow ogee crest, the discharge is given by (Bureau of Reclamation, 1977)

$$Q_c = NCLH^{3/2}$$

---

## Referee Comment (RC2) · Anonymous Referee #2 · 20 Dec 2019

Formally, the quality of the document is unacceptable. The whole document, from the title (!) to the conclusions, is full of grammar and structure mistakes. Only this reason is enough for rejection. This denotes a lack of respect towards the editors and reviewers. The structure of most sentences is totally wrong in English language. It is not a document in English; it is a mere word-by-word translation of a Spanish document.

I am not going to do any list of wrong sentences and grammar mistakes as I think this is not the task of reviewers, but in view of the quality of this document, I suggest the authors to use correction and editing services before submitting documents to quality

journals in the future.

On the following lines there are some comments on the structure of the document and on the technical contents. These comments cannot be considered a thorough detailed revision, as in my opinion the quality of the document is so low, that this task would be nonsense and a loss of time.

1) Starting with the title, from it is impossible to understand what the document is about. In different words, what it says is: "overtopping is the technique used to analyse risk in a Dam while a landslide is taking place upstream of it". Thus, it is totally nonsense!

2) Abstract: Apart from being a series of wrong constructed sentences (was an automatic translator used?) its structure is not adequate to transmit the contents of the work. It starts describing the contents of the document, then it describes the methodology, it goes back to the description of the contents, there is no information on the methods in it, and no description of what type of risk is the document dealing with.

3) Structure of the document. It starts with an introduction (section 1), but only on the case. There is no introduction on the methodology, no antecedents, no background, not a single reference, but, most important, no objectives of the work, and not a line explaining what the structure or contents of the document is. After it, there are several sections, without connections between them or a discursive thread.

4) "2. Landslide" section. There is a description of the landslide, but no information on the sources of information.

5) "4. Geological framework" Section. To start with, there is no section 3. Have the authors re-read their document at least once after writing it? It does not seem so. Same comments s for section 2 on the lack of information on the sources. There is a description of the failure process, but it does not tell the sources or references. The failure study done by the signing authors?.

6) "5. Basis of the study" section. Finally, here, there is some description on the
organization of the document (in section 5!).

7) "5. Background" section. As an example of the quality of the document, it starts with an obvious affirmation: that levees, dams and dikes are located in rivers, but even this is not properly expressed: the authors say that landslide dams are embankment structures, with is not correct.

8) "6. Approach to the Problem" section. I already said it, but the sentences are totally wrong from the point of view of English language. The tiles of the subsections do not correspond with the contents in them. Apart from that, the authors define a "hydrograph produced by the landslide" when probably they mean a "hydrograph produced by the landslide dam break", and they define it as a linear hydrograph, with only one decreasing branch. They say it is triangular, but it is not. But the main point is what is the justification of this hydrograph? The hydrograph should have a raising branch followed by a decreasing one. If they simplify it to a single decreasing discharge branch, they must justify this assumption. They justify the equation of the hydrograph form the hydrograph's figure. But they do not say where does this graph comes from. Thus, the main data used in the work is not justified at all.

In the "discharge section" they present the basic equation of the flow over a rectangular weir and its graph. This is elemental hydraulics. No need of presenting this image.

Section 6.5 is called "Flood routing reviewed". I cannot understand this title.

Section 6.6. They present a numerical development of equations without saying what the purpose of it is. They start saying that the solution is based on the "trapezoidal" rule, without saying what this rule is, or giving any reference. But most importantly, I cannot see the purpose of the whole development. Similar results can be obtained with classic, well known reservoir routing methods (as Pulse, or Modified-Puls methods) that are implemented in many hydrological simulation packages, even free ones, as Hec-HMS.

My revision ends at section 6.6 as I do not see the point of going on with it. Only a final comment on the references: a symptom of the lack of novelty of the document is that there are only three references dated after 2010, and only one of these is after 2015, and in a paper dealing with the topics of dam break, breach formation, reservoir routing, and natural dams, an area where there are many new publications ever year.

Conclusions:

My recommendation is that the paper should be rejected, and with no option to resubmitting it.

I do not think the contents of the work fit with the aims of a serious journal as NHESS. The main work in the document consists of a series of simulations based on simple equations, simple assumptions, and standard equations. There is no novelty on it or any interest for possible readers.
* * *